# Dermoscopic Patterns of Genodermatoses: A Comprehensive Analysis

**DOI:** 10.3390/biomedicines11102717

**Published:** 2023-10-06

**Authors:** Dóra Plázár, Fanni Adél Meznerics, Sára Pálla, Pálma Anker, Klára Farkas, András Bánvölgyi, Norbert Kiss, Márta Medvecz

**Affiliations:** Department of Dermatology, Venereology and Dermatooncology, Faculty of Medicine, Semmelweis University, 1085 Budapest, Hungary; plazar.dora@phd.semmelweis.hu (D.P.); meznerics.fanni.adel@semmelweis.hu (F.A.M.); zakarias.sara@phd.semmelweis.hu (S.P.); anker.palma@med.semmelweis-univ.hu (P.A.); farkas.klara@med.semmelweis-univ.hu (K.F.); banvolgyi.andras@med.semmelweis-univ.hu (A.B.); kiss.norbert@med.semmelweis-univ.hu (N.K.)

**Keywords:** dermoscopy, genodermatosis, BCC, skin cancer, acantholytic, ichthyosis, pseudoxanthoma elasticum, neurofibroma, angiokeratoma, tuberous sclerosis complex

## Abstract

(1) Background: Genodermatoses are a clinically and genetically heterogenous group of inherited skin disorders. Diagnosing inherited skin diseases is a challenging task due to their rarity and diversity. Dermoscopy is a non-invasive, easily accessible, and rapid tool used in dermatology not only for diagnostic processes but also for monitoring therapeutic responses. Standardized terminologies have been published for its proper use, reproducibility, and comparability of dermoscopic terms. (2) Methods: Here, we aimed to investigate dermoscopic features in various genodermatoses by conducting a systematic review and comparing its results to our own findings, data of patients diagnosed with genodermatoses at the Department of Dermatology, Venereology and Dermatooncology, Semmelweis University. (3) Results: Our systematic search provided a total of 471 articles, of which 83 reported both descriptive and metaphoric dermoscopic terminologies of 14 genodermatoses. The literature data were then compared to the data of 119 patients with 14 genodermatoses diagnosed in our department. (4) Conclusion: Dermoscopy is a valuable tool in the diagnosis of genodermatoses, especially when symptoms are mild. To enable the use of dermoscopy as an auxiliary diagnostic method, existing standardized terminologies should be extended to more genodermatoses.

## 1. Introduction

Genodermatoses are a clinically and genetically heterogenous group of inherited skin disorders. These are chronic conditions that present with variable severity of dermatological symptoms and may be associated with extracutaneous manifestations that can have a severe impact on the overall health and quality of life of patients. Diagnosing inherited skin diseases is difficult because these conditions are both rare and diverse. The multistep diagnostic algorithm for inherited skin diseases suggests considering phenotypic features and clinical data, mode of inheritance, target proteins, and genetic variants in the diagnosis of genodermatoses [1].

Dermoscopy is a non-invasive, easily accessible, and rapid tool used in dermatology not only for diagnostic processes but also for monitoring therapeutic responses [2,3,4,5] in the pediatric population [6,7]. To ensure correct use, reproducibility, and comparability of dermoscopic terms, in 2015, Kittler et al. published the standardized terminology as a result of the third consensus conference of the International Society of Dermoscopy. To date, both competitive descriptive and metaphorical terminologies have been used in the dermoscopic literature, but the introduction of further metaphorical terms is not recommended [8]. Despite this, Errichetti et al. argue that these terms can only be applied to skin neoplasms on which the consensus has focused. Hence, they aimed to define dermoscopic terminology and basic parameters in general dermatology to evaluate non-neoplastic dermatoses as well [9]. 

Here, we aimed to investigate dermoscopic features in various genodermatoses based on the literature data and our own findings. We limited ourselves to listing the clinical characteristics of genodermatoses included in our study.

### 1.1. Conditions Affecting the Epidermis, Epidermal Structures, and Appendages

#### 1.1.1. Ichthyoses

Inherited ichthyoses, also referred to as Mendelian Disorders of Cornification (MeDOC), are a genetically and clinically heterogeneous group characterized by hyperkeratosis, diffuse scaling, xerosis, and a variable degree of erythroderma. The severity of symptoms varies widely due to epidermal barrier defects and various disturbances of the terminal differentiation process of keratinocytes. Non-syndromic types of ichthyoses can be distinguished from syndromic ichthyoses. Ichthyosis vulgaris (IV, ASD, OMIM # 146700) is the most frequent type and is caused by autosomal semi-dominant *filaggrin* gene (*FLG*) mutations. The clinical characteristics include fine or prominent scaling over the lower trunk and extremities, palmar hyperlinearity, keratosis pilaris, and frequent association with atopic conditions (Figure 1a,b). X-linked recessive ichthyosis (XLI, XR, OMIM # 308100) occurs almost exclusively in male patients, resulting from steroid sulfatase deficiency, and is caused by deletion of the *STS* gene locus or gene mutation. XLI is clinically characterized by extensive dark brown polygonal scales, but the flexural areas are not involved (Figure 1c,d). Autosomal recessive congenital ichthyosis (ARCI) is both clinically and genetically very heterogeneous, and 70–90% of the cases present at birth with a collodion membrane. Other cases manifest with signs of abnormal cornification until the fourth week of life [10,11,12,13]. On the basis of the inverse relationship between the severity of ichthyosis and erythroderma, the main skin phenotypes are lamellar ichthyosis (LI) and congenital ichthyosiform erythroderma (CIE), although phenotypic overlap can occur. LI (AR, OMIM # 242300) is characterized by generalized large adherent dark scaling with mild erythema (Figure 2c,d); however, CIE (AR, OMIM # 242100) occurs with prominent erythema and fine white scales (Figure 2e,f). Pleomorphic ichthyosis refers to a group of various conditions characterized by a presence of mild congenital ichthyosis with fine scaling that persists after initial skin symptoms during early childhood (Figure 2a,b) [14]. Harlequin ichthyosis (HI, AR OMIM # 242500) is a rare severe often fatal form of ARCI, with thick scale plates and deep fissures (Figure 2g,h) [12].

#### 1.1.2. Dowling–Degos Disease (DDD, AD, OMIM # 179850)

DDD is characterized by slowly progressive reticulate brown-to-black hyperpigmentation typically involving large body folds and flexural areas (Figure 3). Comedone-like follicular papules with hyperkeratosis, hypopigmented lesions, and pitted perioral scars can usually develop during adulthood. Mutations in genes such as *KRT5*, *POFUT1*, *POGLUT1*, and *PSENEN* affecting melanosome transfer, melanocyte, and keratinocyte differentiation are affected in the pathogenesis of DDD [15].

#### 1.1.3. Palmoplantar Keratodermas

Hereditary palmoplantar keratodermas are a heterogeneous group of keratinization disorders marked by excessive thickening of the epidermis of palms and soles. The clinical morphology of hyperkeratosis may be diffuse, focal/striate, or papular/punctate (Figure 4c,d). Mutation analysis is necessary to define the exact type of PPK. Diffuse epidermolytic PPK (EPPK, AD, OMIM # 144200) is the most common diffuse PPK with epidermolytic changes in suprabasal keratinocytes due to mutations in *KRT9* and rarely in *KRT1* genes. EPPK patients develop confluent fissured brown/yellow hyperkeratosis affecting only palmoplantar surfaces with an erythematous edge (Figure 4a,b). Mutations in the *AAGAB* gene result in punctate PPK (PPPK, AD, OMIM # 148600) [16].

#### 1.1.4. Erythrokeratodermia Variabilis et Progressiva (EKVP) 

Erythrokeratodermia variabilis et progressiva is a clinically and genetically heterogeneous group of inherited disorders characterized by hyperkeratotic plaques and transient erythematous patches (Figure 5). Mutations affect *GJB3* (EKVP1, AD or AR, OMIM # 133200), *GJB4* (EKVP2, AD, OMIM # 617524), and *GJA1* (EKVP3, AD, OMIM # 617525), encoding different types of connexins and four other genes, as well as other plasma membrane components [17,18,19,20,21].

#### 1.1.5. Darier Disease (DD, Keratosis Follicularis, AD, OMIM # 124200)

DD is characterized by loss of adhesion between epidermal cells and abnormal keratinization, caused by mutations of the *ATP2A2* gene, which encodes an endoplasmic reticulum calcium pump (sarco/endoplasmic reticulum ATPase type 2 (SERCA2)). It usually manifests in small keratotic papules or plaques predominantly in the seborrheic areas such as the chest, back, and also the face (Figure 6). Nail abnormalities, such as longitudinal erythronychia and leukonychia (Figure 7a–d), acral lesions, mucous membrane changes, and neuropsychiatric abnormalities may also appear [22].

#### 1.1.6. Hailey–Hailey Disease (HHD, Benign Chronic Pemphigus, AD, OMIM # 1696000)

HHD is caused by mutations of the ATPase secretory pathway Ca^2+^ transporting 1 gene, *ATP2C1*. It typically manifests in painful erosions, fissures, vesicopustules, and scaly erythematous plaques classically involving the intertriginous areas such as the axilla, sub-mammary area, groin, and perineum, often in a symmetrical distribution (Figure 8). Longitudinal leukonychia may also appear (Figure 7e,f) [23,24].

#### 1.1.7. Monilethrix (MNLIX, AD, OMIM # 158000)

MNLIX is characterized by hair shaft dysplasia and fragility, resulting in hypotrichosis, especially in the occipital region, or alopecia of variable severity (Figure 9a). Microscopic examination of the hair shaft reveals periodic elliptical nodes and intermittent internodal constrictions leading to characteristic “beaded ribbon” appearance of the hair (Figure 9b). AD forms are associated with mutations in hair keratin genes (*KRT81*, *KRT83*, and *KRT86*) [25].

### 1.2. Connective Tissue Disorder

#### Pseudoxanthoma Elasticum (PXE, AR, OMIM # 264800)

Mutations of the ATP-binding cassette subfamily C gene, *ABCC6,* cause calcification and fragmentation of elastic fibers in the skin, blood vessels, and the retina. It results in increased laxity and loss of elasticity of the skin, arterial insufficiency, and retinal hemorrhages. Dermatological examination reveals multiple coalescing soft yellowish papules with a cobblestone appearance that are symmetrically distributed on the neck, nape, and other flexural areas of the body (Figure 10) [26,27].

### 1.3. Lysosomal Storage Disorder

#### Fabry Disease (FD, XL, OMIM # 301500)

FD is an X-linked inherited disorder of the glycosphingolipid metabolism, caused by a variety of mutations in the *alpha-galactosidase A* gene (*GLA*), resulting in progressive accumulation of globotriaosylceramide, especially in endothelial cells, causing multi-organ damage. Angiokeratoma corporis diffusum universale is a distinctive cutaneous manifestation of FD. It is characterized by the presence of widespread angiokeratomas typically located in the bathing suit distribution between the navel and the knees (Figure 11) [28,29,30].

### 1.4. Neurocutaneous Conditions

#### 1.4.1. Neurofibromatosis Type 1 (NF1, von Recklinghausen’s Disease, AD, OMIM # 162200)

NF1 is characterized by multiple cutaneous neurofibromas (Figure 12) and café-au-lait macules (CALMs, Figure 13a,b), axillar, inguinal or diffuse freckling, and less often juvenile xanthogranuloma or nevus anemicus. It is caused by mutations of the *NF1* gene leading to dysfunction of the tumor suppressor NF1 protein (neurofibromin) [31].

#### 1.4.2. Tuberous Sclerosis Complex (TSC, AD, OMIM # 191100)

TSC is caused by mutations of tumor suppressor genes *TSC1* and *TSC2*, resulting in hyperactivation of the mTOR signaling pathway. It manifests in hamartomas that may affect multiple organs such as skin, heart, lungs, central nervous system, and kidneys. Cutaneous manifestations are hypopigmented “ash-leaf” (Figure 13c,d) and smaller roundish “confetti” macules, facial angiofibromas (Figure 14a,b), shagreen patches (connective tissue nevus, Figure 14c,d), and ungual or periungual Koenen fibromas [32].

#### 1.4.3. Basal Cell Nevus Syndrome (BCNS or Nevoid Basal Cell Carcinoma Syndrome (NBCCS) or Gorlin–Goltz Syndrome (GGS), AD, OMIM # 109400)

Mutations in the tumor suppressor gene *PTCH1*, and in other modifier *PTCH2* and *SUFU* genes, present with multiple early-onset basal cell carcinoma (BCC, Figure 15a,b), palmar and plantar pits (Figure 15c,d), multiple odontogenic keratocysts, and skeletal abnormalities, and are also alternately associated with a broad spectrum of developmental anomalies and neoplasms [33].

### 1.5. Other Syndromes Affecting the Skin

#### 1.5.1. CYLD Cutaneous Syndrome ((CCS) including Brooke–Spiegler Syndrome (BRSS), AD, OMIM # 605041; Familial Cylindromatosis (FC), OMIM # 132700; Multiple Familial Trichoepitheliomas (MFT), OMIM # 601606)

CCS is an inherited skin adnexal tumor syndrome caused by mutations in the *CLYD* gene. It usually manifests in multiple cylindromas, trichoepitheliomas, and spiradenomas located on the head and neck (Figure 16). The size and the number of these appendage tumors typically increase throughout life [34].

#### 1.5.2. Noonan Syndrome with Multiple Lentigines (NSML)/Noonan Syndrome 1 ((NS1), AD, OMIM # 163950)/LEOPARD Syndrome 1 ((LPRD1) or Multiple Lentigines Syndrome, AD, OMIM # 151100)

NSML is mainly caused by defined mutations in the *PTPN11* gene. It is characterized by multiple cutaneous lentigines, CALMs, hypertrophic cardiomyopathy and ECG abnormalities, short stature, pectus deformity, dysmorphic facial features, and sensorineural hearing loss [35]. Skin lesions include two types of spots. Lentigines are 1–2 mm sized, brown to black colored macules, and increase in number until puberty. Café noir spots are darker and larger than lentigines, up to 5 cm in diameter (Figure 17) [36].

## 2. Materials and Methods

### 2.1. Systematic Review

Our results are reported according to the guidelines of the PRISMA (Preferred Reporting Items for Systematic Reviews and Meta-Analyses) 2020 Statement [37]. We registered the review protocol on PROSPERO under registration number *CRD42023452448*.

A literature search was conducted on 8 August 2023, using Pubmed, Embase, and Cochrane (CENTRAL) databases to identify eligible records. The search key “(dermoscopy OR dermatoscopy) AND (“Darier disease” OR “Hailey-Hailey disease” OR monilethrix OR “Fabry disease” OR “Dowling-Degos” OR “tuberous sclerosis complex” OR “neurofibromatosis” OR “basal nevoid cell syndrome” OR “Gorlin Goltz” OR “Gorlin syndrome” OR “pseudoxanthoma elasticum” OR ichthyosis OR Harlequin OR “palmoplantar keratoderma” OR “erythrokeratodermia variabilis et progressiva” OR “Noonan syndrome” OR “LEOPARD syndrome” OR “trichoepithelioma” OR “Brooke-Spiegler” OR “shagreen patch” OR “cafe au lait”)” was applied. No language or other restrictions were imposed during the search process. Original articles, case reports, short communications, correspondences, and letters describing the dermoscopic features of skin lesions of Darier disease, Hailey–Hailey disease, Dowling–Degos disease, pseudoxanthoma elasticum, tuberous sclerosis complex, neurofibromatosis type 1, LEOPARD syndrome, Fabry disease, basal nevoid cell syndrome, ichthyosis vulgaris, autosomal recessive ichthyosis, lamellar ichthyosis, annular epidermolytic ichthyosis, and Brooke–Spiegler syndrome were included. Language articles not in English were excluded.

Selection and data extraction were conducted by two independent authors using EndNote X9 (Clarivate Analytics, Philadelphia, PA, USA) and Excel spreadsheet (Office 365, Microsoft, Redmond, WA, USA).

The quality assessment was performed using the JBI Critical Appraisal tool for case reports and case series [38,39].

### 2.2. Descriptive Study

The prospective dermoscopic imaging study was carried out in the Department of Dermatology, Venereology and Dermatooncology, Semmelweis University between September 2020 and January 2023. The study was conducted according to the declaration of Helsinki. A total of 119 patients with 14 different inherited disorders were evaluated. Patients with the previously established diagnosis of genodermatosis were included. Exclusion criteria were diagnoses of other skin diseases (e.g., skin infections) that may interfere with dermoscopic features. Diagnosis was confirmed based on the current diagnostic guideline for each disease. Patients gave written informed consent to this study. Demographic data, such as age, gender, and the type of genodermatosis, were documented. All patients underwent detailed clinical examinations. Clinically relevant skin lesions were selected for dermoscopic analysis. Clinical and dermoscopic images were captured. Dermoscopy was performed using Illuco IDS-1100C (Illuco Corporation Ltd., Gunpo, Republic of Korea) and Heine dermatophot (10-fold magnification, Heine Optotechnik GMBH & CO. KG., Gilching, Germany) with an optional polarized light source. All authors evaluated the dermoscopic images. Standardized terminologies and processes suggested by Kittler et al. and Errichetti et al. were applied, with the exception of neurofibromas, where the terms used by Duman et al. were used. Onychoscopic and trichoscopic findings were based on case reports and reviews. Comparison of our own findings to those reported in the literature was carried out.

## 3. Results

Our systematic search provided a total of 471 articles; we identified 74 eligible studies by title, abstract, and full-text selection [23,24,26,27,30,40,41,42,43,44,45,46,47,48,49,50,51,52,53,54,55,56,57,58,59,60,61,62,63,64,65,66,67,68,69,70,71,72,73,74,75,76,77,78,79,80,81,82,83,84,85,86,87,88,89,90,91,92,93,94,95,96,97,98,99,100,101,102,103,104,105,106,107,108], and 9 additional studies by citation searching [7,109,110,111,112,113,114,115,116]. The selection process is summarized in Figure 18 (PRISMA). 

### 3.1. Systematic Review

Characteristics of studies included for the systematic review are detailed in Table 1. 

We summarized the findings of the studies included in the systematic review in Table 2.

The results of the risk of bias assessment of the studies are detailed in Table A1 and Table A2 in the Appendix A.

### 3.2. Descriptive Study

The number of patients, analyzed areas or lesions, and the affected areas for each disease are summarized in Table 3.

The dermoscopic analysis of our results following the terminology of Errichetti et al. and Kittler et al. are summarized in Table 4 and Table 5. Both descriptive and metaphoric terminologies are applied. Metaphoric terms are printed in bold and italics.

The trichoscopic and onychoscopic findings are summarized in Table 6.

## 4. Discussion

Genodermatoses are a large group of inherited skin diseases whose diagnosis is challenging due to their rarity and clinical and genetic diversity [117].

Given the dynamical development of preclinical and clinical studies in various genodermatoses in recent years to assess the applicability of different targeted therapies (gene, cell-based, protein therapy) and symptom-relief therapies (repurposed and new orphan drugs), it would be important to have non-invasive diagnostic tools for objective assessments of skin conditions.

Dermoscopy is one of the useful non-invasive tools in the diagnosis and follow-up of many dermatoses such as inherited rare skin diseases. There are competing descriptive and metaphoric terminologies in the literature. Metaphoric terms may be illustrative and memorable; however, sometimes they may also present a level of ambiguity and lack of clarity, potentially leading to difficulties in everyday clinical practice. Descriptive terminology is clear and logical but may have limitations when describing complex dermoscopic structures. 

Standardized dermoscopic terminology by Kittler et al. can be used properly to analyze lesions in FD, NF1, BCNS, NSML, and CCS. Expanded terminology on general dermatology by Errichetti et al. may include parameters describing ichthyoses, PPKs, EKVP, DD, HHD, DDD, PXE, and TSC. For the trichoscopy of MNLIX and onychoscopic analysis, we applied the terms introduced in case reports and review articles.

Dermoscopy is useful for making a diagnosis, especially when skin manifestations are less pronounced. In our results, it was applicable for detecting characteristic papules in one mild case of DD, visualizing an erythematous edge in a newborn with EPPK and trichoepitheliomas in CCS, differentiating angiokeratomas from hemangiomas in FD, and choosing the proper area for biopsy in a mild case of PXE. Dermoscopy may also enhance monitoring of disease activity and accurate follow-up of treatment response. Errichetti et al. successfully used dermoscopy in psoriasis. According to their results, it was useful for following therapy response, detecting steroid-induced skin atrophy by visualizing characteristic linear vessels, and disease recurrence [118]. In our cases, steroid-induced skin atrophy could be seen in patients with HHD and DD. In addition, with the use of dermoscopy, we monitored the efficiency of topical therapy for adenoma sebaceum (angiofibroma) in TSC. In our clinical practice, we used dermoscopy for the follow-up of patients with BCNS or NSML to detect potential skin tumors. 

Here, we expanded the literature on dermoscopic analysis of many genodermatoses, including nail findings as well. According to recommendations, no new metaphoric terms were added to the literature. To our knowledge, this is the first report on the use of dermoscopy in EPPK, EKVP, and some ARCI such as LI, pleomorphic, and Harlequin ichthyosis. Dermoscopy of PPK and shagreen patch in TSC were described in only one case report of both diseases, including dermoscopic images as well. Our results were similar in dermoscopic features of PPKs; however, in shagreen patch, we described white/light yellow structureless areas with vessels that differed from the findings reported in the literature (reddish brown strands with white lines with a cobblestone appearance) [119]. This may be because of the different ethnicities of the two patients. 

To use dermoscopy as an auxiliary diagnostic tool in the diagnosis of genodermatoses, existing standardized terminologies (both descriptive and metaphoric) should be expanded to more phenotypes of genodermatoses.

## Figures and Tables

**Figure 1 biomedicines-11-02717-f001:**
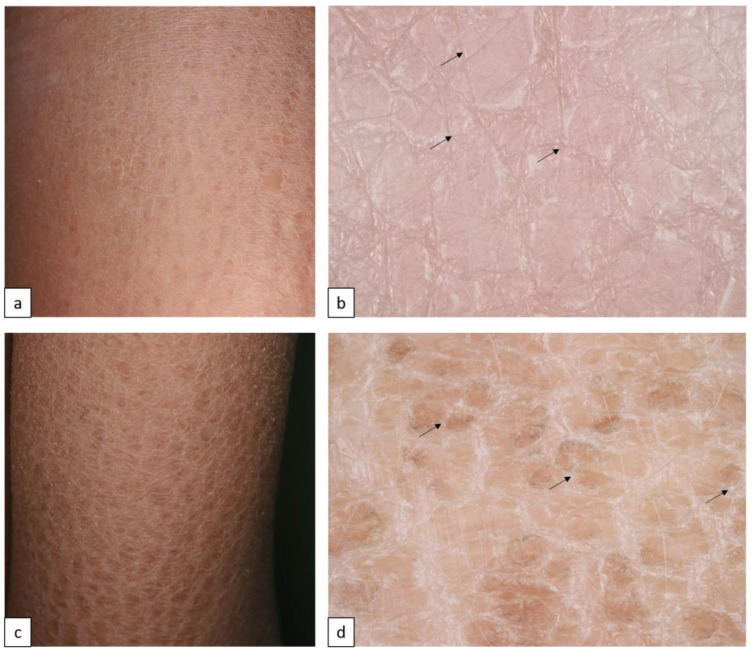
Common forms of inherited ichthyoses. Ichthyosis vulgaris is characterized by fine white or light gray scales (**a**). Dermoscopy shows a criss-cross pattern of fine white scales ((**b**), arrows). X-linked recessive ichthyosis manifests in large firmly attached brown rhomboid scales (**c**). Dermoscopy reveals a mosaic pattern of brown structures with space in between ((**d**), arrows).

**Figure 2 biomedicines-11-02717-f002:**
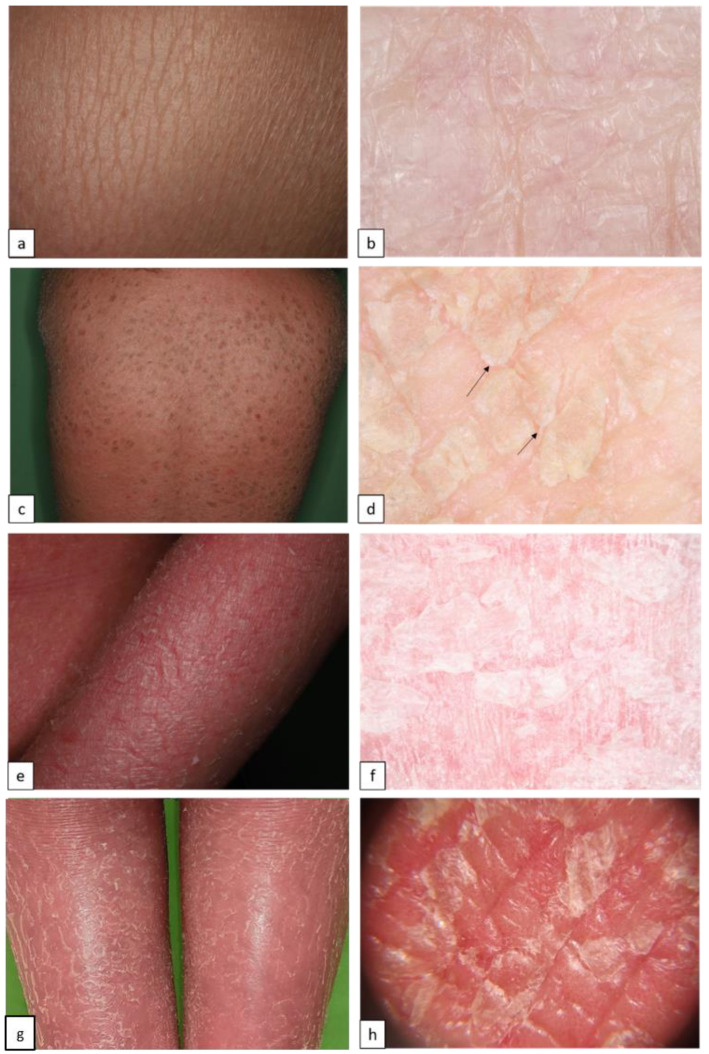
Autosomal recessive congenital ichthyoses. Pleomorphic ichthyosis (**a**) manifests in fine white scales (**b**). Generalized large brown lamellar scaling with mild erythema in lamellar ichthyosis (**c**). Dermoscopy shows quadrilateral yellow/brown scales (**d**), arrows arranged in rhomboid pattern (**d**). Diffuse variable size of polygonal white or light gray scales and background erythema in congenital ichthyosiform erythroderma (**e**,**f**). Clinical and dermoscopic images of Harlequin ichthyosis reveal extensive background erythema, dotted vessels, and white scales in variable size and form (**g**,**h**).

**Figure 3 biomedicines-11-02717-f003:**
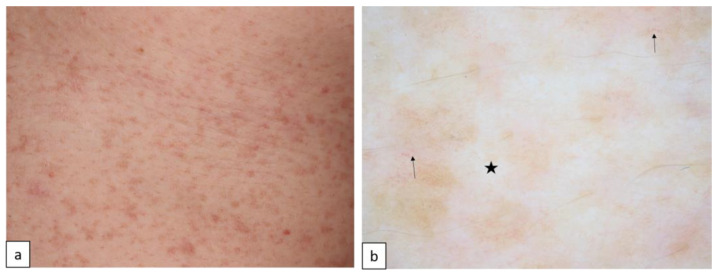
Dowling–Degos disease (**a**). Dermoscopy shows yellow/brown structureless areas, white globules coalescing into lines ((**b**), star), and linear vessels ((**b**), arrows).

**Figure 4 biomedicines-11-02717-f004:**
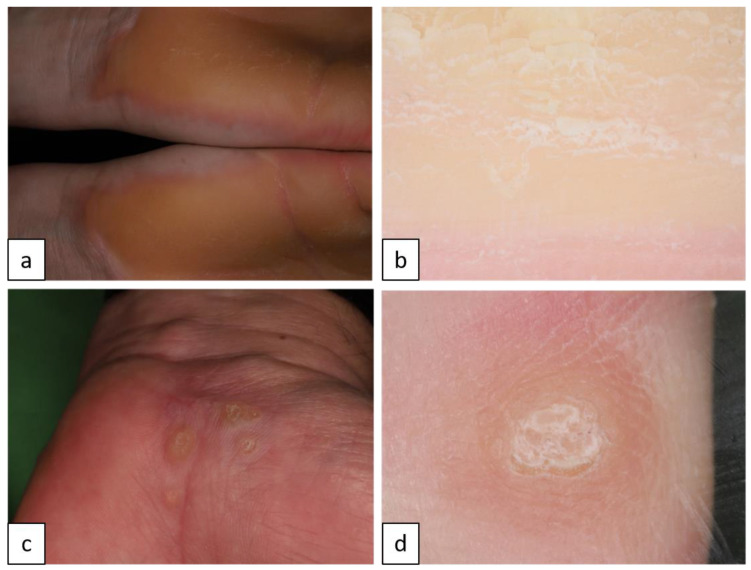
Diffuse epidermolytic palmoplantar keratoderma appears as yellow/white scales, fissures, and epidermolytic hyperkeratosis (**a**). Under dermoscopy, white/yellow hyperkeratosis, fissures, and homogenous erythematous areas can be seen (**b**). Punctate palmoplantar keratoderma of the palms (**c**). Dermoscopy reveals multiple round yellow areas with hyperkeratosis and white/yellow scales (**d**).

**Figure 5 biomedicines-11-02717-f005:**
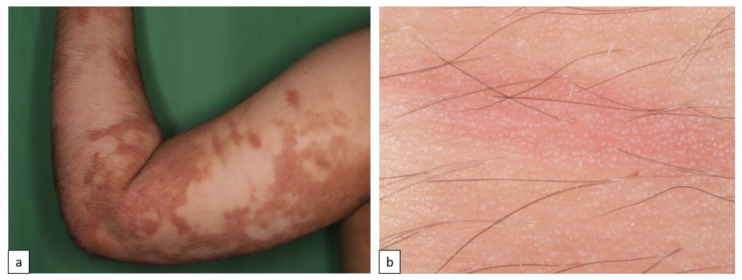
Erythrokeratodermia variabilis et progressiva. Confluent hyperkeratotic plaques and erythematous patches affect the arm (**a**). Dermoscopy shows brown lines, erythema, and white hyperkeratotic globules (**b**).

**Figure 6 biomedicines-11-02717-f006:**
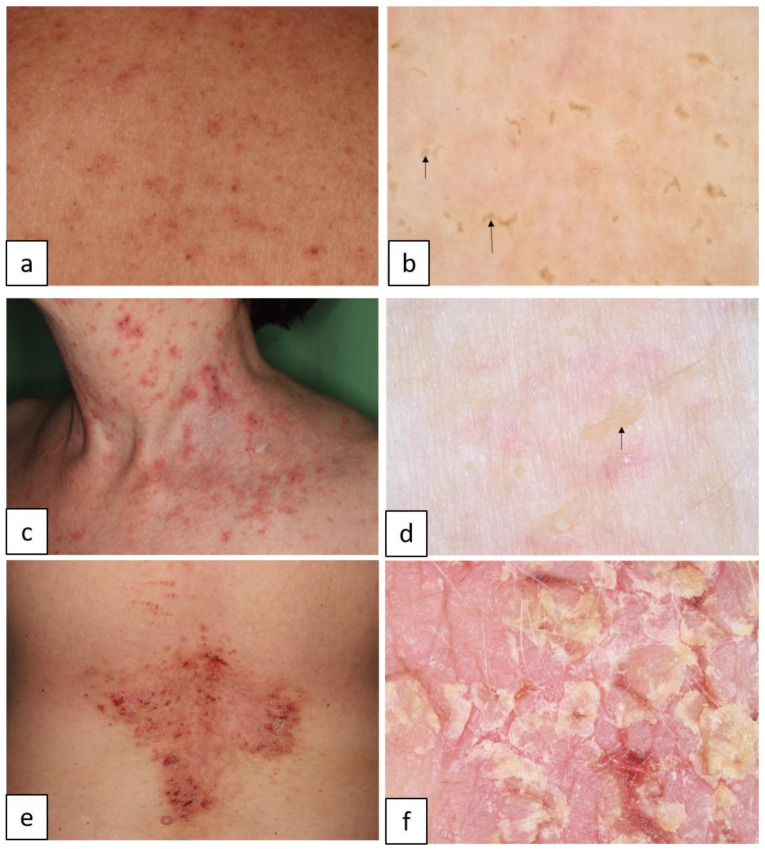
Darier disease. Discrete brownish erythematous hyperkeratotic papules and plaques on the neck (**a**) and on the back (**c**) and severe symptoms affecting the lumbosacral region (**e**). Dermoscopic image of yellow/brown areas ((**b**,**d**), arrows) has a polygonal shape, surrounded by white halo representing the acantholytic epidermis. Under dermoscopy, plaque-type lesions appear as erosions, erythematous structureless areas, and yellow/white scales (**f**).

**Figure 7 biomedicines-11-02717-f007:**
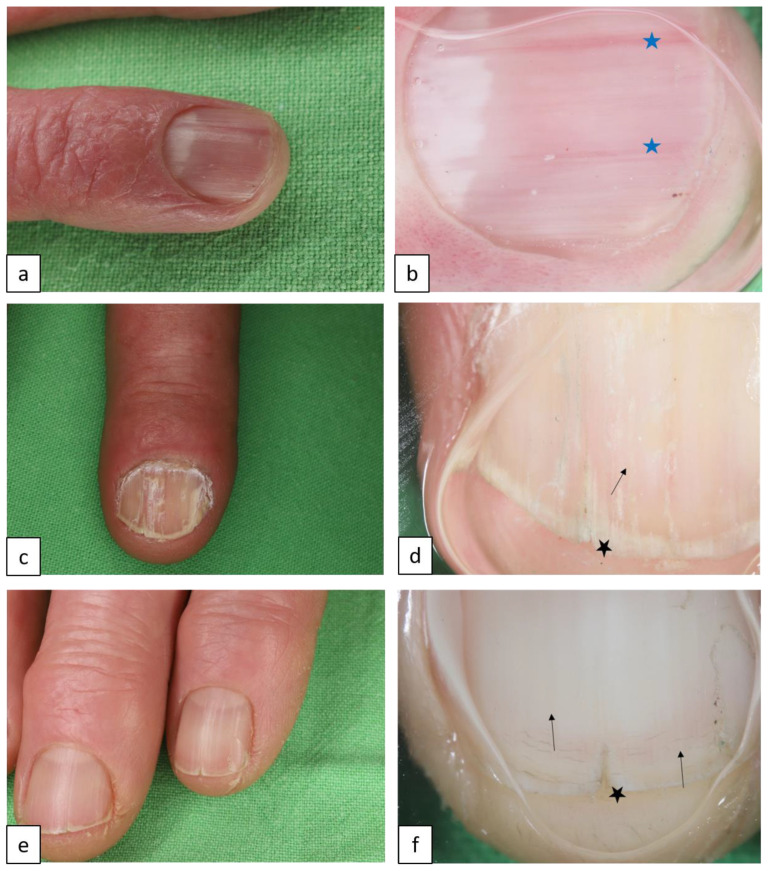
Nail findings in acantholytic genodermatoses (**a**,**c**,**e**). Onychoscopy reveals red ((**b**), blue stars), and white longitudinal bands ((**d**,**f**), arrows), and V-shaped nick ((**d**,**f**), black stars) in Darier disease (**a**–**d**) and in Hailey–Hailey disease (**e**,**f**).

**Figure 8 biomedicines-11-02717-f008:**
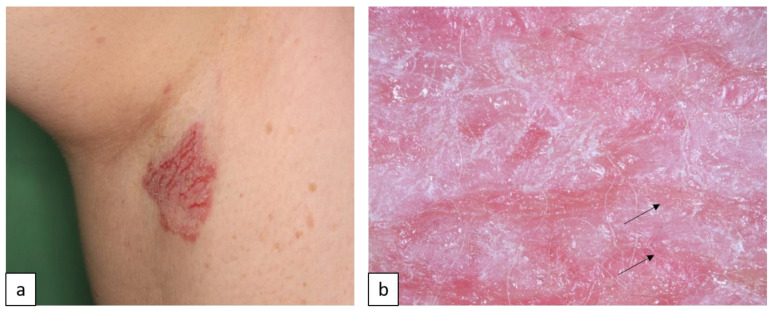
Hailey–Hailey disease. Erythematous plaques with erosions and fissures in the axilla (**a**). Dermoscopy shows white structureless areas separated by parallel lines and erosions ((**b**), arrows).

**Figure 9 biomedicines-11-02717-f009:**
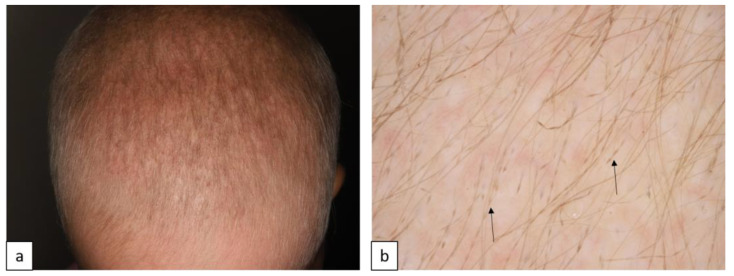
Diffuse hypotrichosis and coarse hair in a patient with monilethrix (**a**). Trichoscopy reveals periodic thinning of the hair shaft leading to characteristic beaded appearance ((**b**), arrows).

**Figure 10 biomedicines-11-02717-f010:**
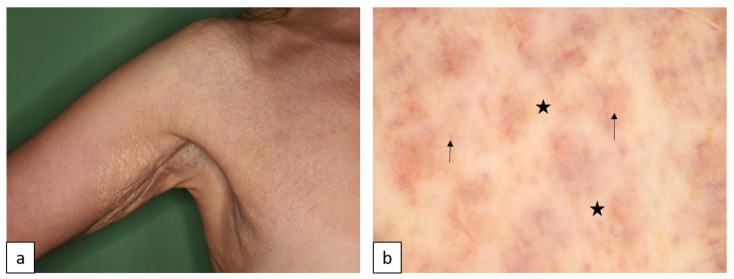
Pseudoxanthoma elasticum. Multiple and coalescing asymptomatic soft yellow papules in the axilla (**a**). Dermoscopy shows yellow/white globules that coalesce into reticular strands ((**b**), stars) on a light purple background with superficial linear vessels ((**b**), arrows).

**Figure 11 biomedicines-11-02717-f011:**
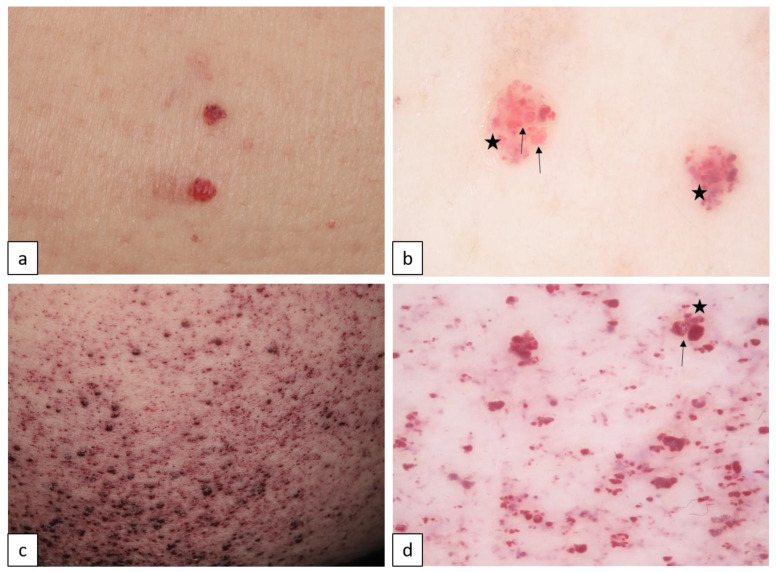
Solitary and multiple angiokeratomas in Fabry disease (**a**,**c**). Dermoscopy in both cases reveals well-demarcated round lacuna ((**b**,**d**), arrows), representing dilated dermal vessels and a whitish veil ((**b**,**d**), stars) as the sign of epidermal hyperkeratosis.

**Figure 12 biomedicines-11-02717-f012:**
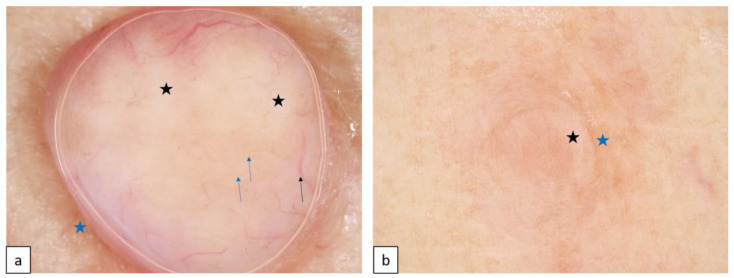
Dermoscopy of neurofibromas in neurofibromatosis type 1 (**a**,**b**) shows pink/red structureless areas, linear vessels ((**a**), black arrow), scar-like areas ((**a**,**b**), black stars), fingerprint-like structures ((**a**), blue arrows), and peripheral halo of brown pigmentation ((**a**,**b**), blue star).

**Figure 13 biomedicines-11-02717-f013:**
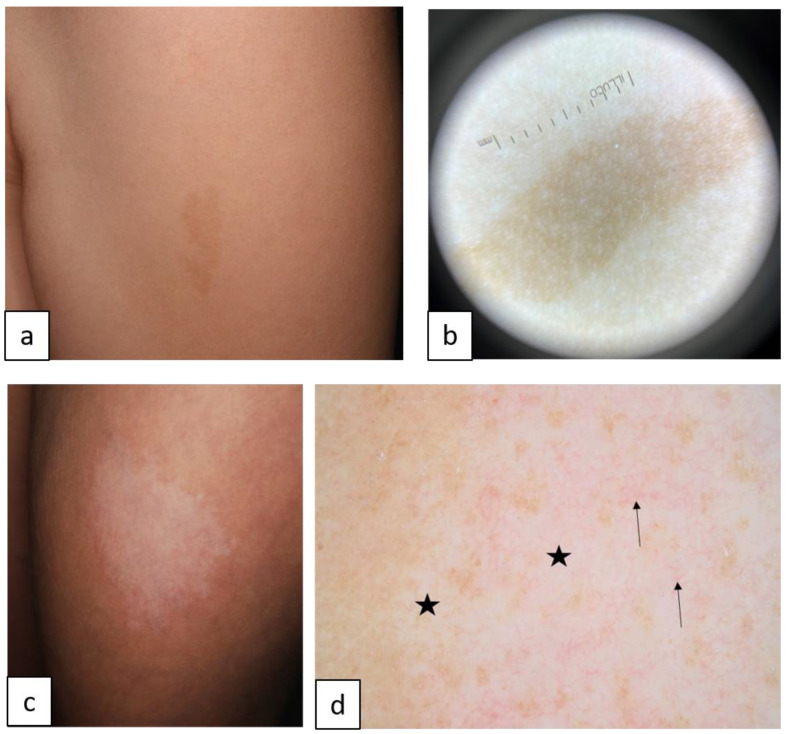
Hypo- and hyperpigmentation in two different neurocutaneous syndromes. Café-au-lait macules in neurofibromatosis type 1 (**a**,**b**). Dermoscopy reveals homogenous brown pigmentation with perifollicular hypopigmentation or reticular pattern of brown pigmentation (**b**). Ash leaf macules on the thigh in tuberous sclerosis complex (**c**,**d**). Under dermoscopy, white globules coalesce into reticulated lines (stars) with feathery irregular border and linear curved vessels ((**d**), arrows).

**Figure 14 biomedicines-11-02717-f014:**
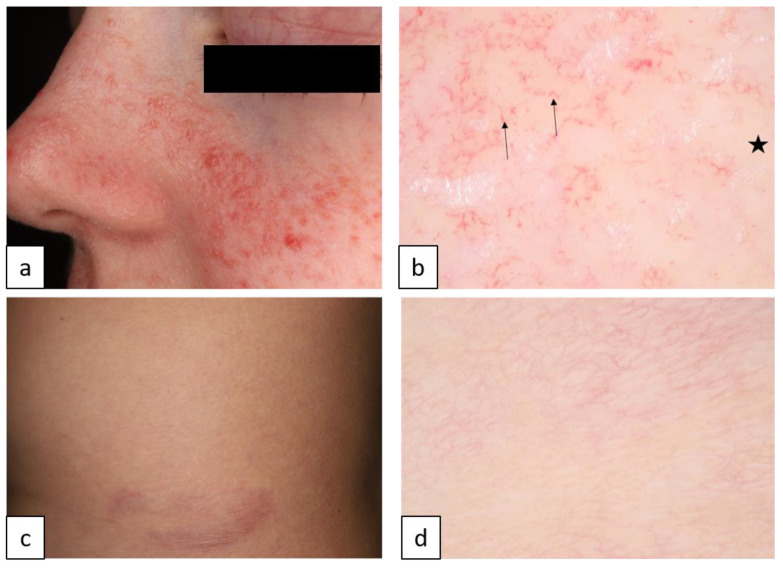
Tuberous sclerosis complex. Adenoma sebaceum (angiofibroma) on the face (**a**). Dermoscopy shows yellow/white dots and globules ((**b**), stars), white structureless areas, and various forms of vessels ((**b**), arrows). Dermoscopic image of shagreen patch on the trunk (**c**) reveals white/yellow structureless areas and reticular vessels (**d**).

**Figure 15 biomedicines-11-02717-f015:**
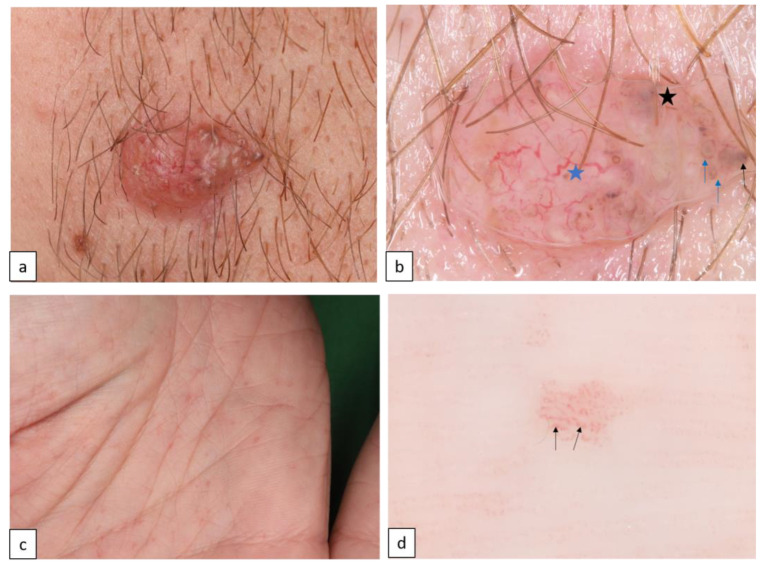
Basal cell nevoid syndrome. Basal cell carcinoma on the face (**a**). Dermoscopy reveals arborizing vessels (blue star, (**b**)), concentric structures ((**b**), blue arrows), grey dots ((**b**), black arrow) and maple-leaf like structures ((**b**), black star). Palmar pits (**c**). Under dermoscopy, pinkish areas appear as red dots in parallel lines ((**d**), arrows).

**Figure 16 biomedicines-11-02717-f016:**
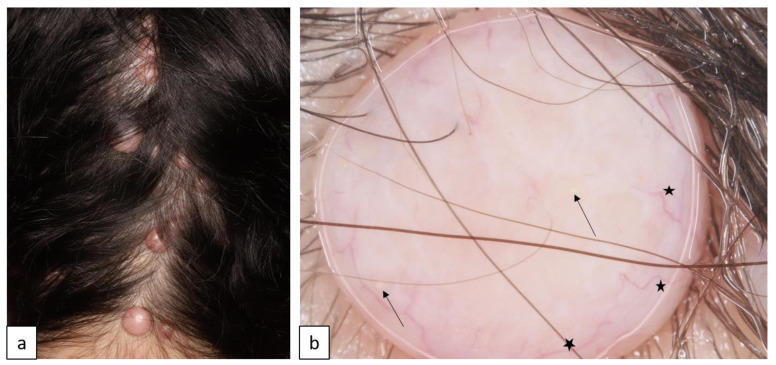
Trichoepitheliomas on the scalp in CYLD cutaneous syndrome (**a**). Dermoscopy reveals milia-like cysts ((**b**), arrows), pink/white background, arborizing vessels ((**b**), stars).

**Figure 17 biomedicines-11-02717-f017:**
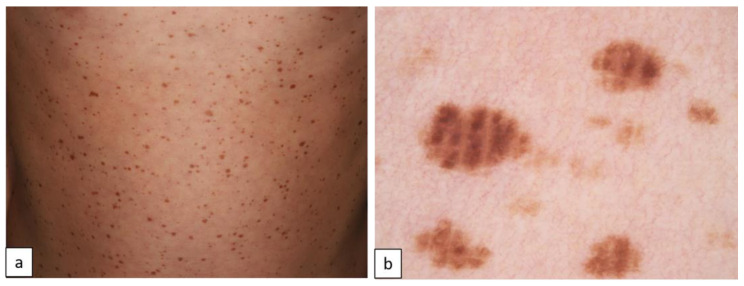
Clinical picture of multiple lentigines and cafe noir spots in Noonan syndrome with multiple lentigines (**a**). Dermoscopy reveals brown pigmentation in a cobblestone pattern (**b**).

**Figure 18 biomedicines-11-02717-f018:**
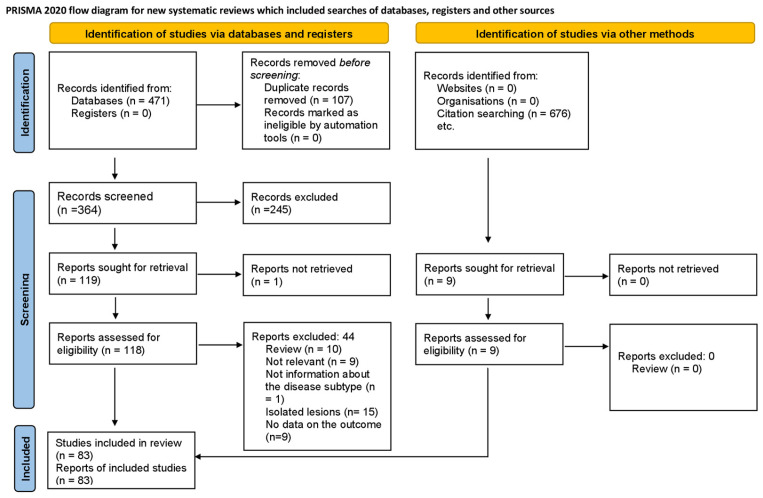
PRISMA Flow Diagram of the screening and selection process.

**Table 1 biomedicines-11-02717-t001:** Characteristics of studies included in the systematic review.

First Author	Year	Study Type	Relevance	Number of Patients
Vázquez-López et al. [115]	2004	brief report	DD	5
Lacarrubba et al. [27,74]	2015, 2017	case reports	DD, PXE	2, 2
Errichetti et al. [58,59,116]	2016, 2023	letter, case report, observational study	DD	11, 1, 22
Oliviera et al. [83,84]	2018, 2019	original article, letter	HHD, DD	8, 6
Peccerillo et al. [85]	2020	case report	DD	1
Siemianowska et al. [94]	2021	case report	DD	1
Dhanaraj et al. [55]	2022	case report	DD	1
Balić et al. [42]	2022	letter	DD	2
Kelati et al. [71]	2017	short communication	HHD	1
Chauhan et al. [49,50,51]	2018, 2019 2021	case reports, correspondence	HHD, PXE	1, 1, 1
Vasani and Save [114]	2019	letter	HHD	1
Narkhede et al. [80]	2021	original article	HHD	2
Ankad et al. [40,109]	2017, 2023	original article, correspondence	TSC, HHD	4, 23
Bel et al. [23,24]	2010, 2014	case reports	HHD	3, 10
Massone et al. [78]	2008	correspondence	DDD	1
Geissler et al. [62]	2011	case report	DDD	1
Dabas et al. [53]	2020	case report	DDD	3
Nirmal et al. [110]	2016	correspondence	DDD	1
Papadopoulou et al. [111]	2022	case report (minireview)	DDD	2
Coco et al. [52]	2019	correspondence	DDD	2
Singh et al. [26]	2017	case report	PXE	1
Kawashima et al. [70]	2018	concise report	PXE	2
Elmas et al. [57]	2021	letter	PXE	1
Salas-Alanis et al. [90]	2019	letter	PXE	1
Berthin et al. [46]	2019	letter	PXE	16
Farkas et al. [60]	2021	original article	PXE	5
Jha et al. [67]	2018	case reports	PXE	1
Vishwanath et al. [101,102]	2019, 2020	case reports	PXE	1, 2
Persechino et al. [86]	2019	letter	PXE	1
Nasca et al. [81]	2016	case report	PXE	1
Anker et al. [30]	2023	article	FD	26
Jindal et al. [69]	2021	letter	TSC	4
Behera et al. [45]	2017	letter	TSC	1
Jimenez-Cauhe et al. [68]	2020	case report	TSC	1
Sechi et al. [91]	2019	brief report	TSC	7
Duman et Elmas [56]	2015	letter	NF1	5
Luk et al. [77]	2014	original article	NF1	4
Gajjar et al. [7]	2019	observational study	MNLIX, TSC, IV, XLI, LI	2, 6, 8
Silverberg et al. [95]	2011	clinical trial	IV	2
Saini et al. [113]	2021	letter	IV, DDD	1
Liang et al. [75]	2020	article	AEI	2
Takeda et al. [97]	2018	case report	ARCI-LI	1
Xue et al. [104]	2019	original contribution	PPK	1
Kolm et al. [72]	2006	case report	BCNS	5
Casari et al. [47]	2017	brief report	BCNS	1
Moreira et al. [79]	2015	case report	BCNS	1
Tiodorovic et al. [99,100]	2010, 2015	case report	BCNS, CCS	1, 2
Jarrett et al. [65,66]	2009, 2010	case reports	CCS, BCNS	2, 4
Sławińska et al. [96]	2018	letter	BCNS	1
Yorulmaz et al. [105]	2017	case report	BCNS	1
Tiberio et al. [98]	2011	case report	BCNS	2
Kosmidis et al. [73]	2023	case report	BCNS	1
Feito-Rodríguez et al. [61]	2009	case report	BCNS	1
Sharma S. et al. [92]	2018	case report	CCS	1
Navarrete-Dechent et al. [82]	2016	case report	CCS	1
Wibowo et al. [103]	2023	case report	CCS	1
Pinho et al. [112]	2015	case report	CCS	2
Ardigo et al. [41]	2007	case report	CCS	4
Banuls et al. [44]	2018	letter	NSML	3
Guliani et al. [63]	2018	case report	NSML	1
Rajamohanan et al. [87]	2020	case report	MNLIX	3
Jain et al. [64]	2010	case report	MNLIX	2
Liu et al. [76]	2008	correspondence	MNLIX	1
Baltazard et al. [43]	2017	case report	MNLIX	1
Sharma VK et al. [93]	2016	letter	MNLIX	1
Rakowska et al. [88,89]	2007, 2008	case reports	MNLIX	1, 1
Zaouak et al. [106]	2019	case report	MNLIX	1
Castañeda-Yépiz et al. [48]	2018	letter	MNLIX	1
De Oliveira et al. [54]	2015	case report	MNLIX	1
Zhi et al. [107]	2018	case report	MNLIX	1
Zhou et al. [108]	2022	case report	MNLIX	3

DD—Darier disease; PXE—pseudoxanthoma elasticum; HHD—Hailey–Hailey disease; DDD—Dowling–Degos disease; FD—Fabry disease; TSC—tuberous sclerosis complex; NF1—neurofibromatosis type 1; MNLIX—monilethrix; IV—ichthyosis vulgaris; XLI—X-linked recessive ichythyosis; LI—lamellar ichthyosis; AEI—annular epidermolytic ichthyosis; ARCI-LI—autosomal recessive congenital ichthyoses-lamellar ichthyosis; PPK—palmoplantar keratoderma; BCNS—basal cell nevoid syndrome; CCS—CYLD cutaneous syndrome ((BRSS) Brooke–Spiegler syndrome); NSML—NSML Noonan syndrome with multiple lentigines.

**Table 2 biomedicines-11-02717-t002:** Dermoscopic findings of genodermatoses of the studies included in the systematic review.

Genodermatosis	Dermoscopic Findings Described in the Literature
Ichthyosis vulgaris	- prominence of linear dermatoglyphic patterning, raised or ragged keratinocyte borders, background erythema, and presence of dull sheen [95] - criss-cross pattern of fine white scale [7,113]
X-linked recessive ichthyosis	rhomboid/mosaic pattern of brown structures with space in between [7]
ARCI-lamellar ichthyosis	- multiple large keratotic plugs in the cristae cutis, highly accentuated sulci cutis [97] - quadrilateral brownish structures with fine white scale arranged in lamellar pattern [7]
Annular epidermolytic ichthyosis	white scales and diffuse punctate hemorrhages [75]
Dowling–Degos disease	- multiple hyperpigmented brownish spots with a regular [78] or reticular pattern [52] characterized by a coarse grid of brown lines over a diffuse light brown background, follicular plugging, and inclusion cysts [62] - brownish projections around a hypopigmented center [110] - brown pigmentation in Chinese letter pattern/irregular star shape, central brown follicular plugs, and comedones [53] - verrucous papules and plaques [111]
Palmoplantar keratoderma	scales and pigmentation, thickened yellow stripes stratum corneum with punctate bleeding [104]
Darier disease	- variable vascular structures (red dots, red lines, or erythema), dilated openings with raised or flat borders, and central brown or yellowish hyperkeratotic plugs [115] - polygonal, starlike, or roundish-oval-shaped yellowish/brownish areas of various sizes surrounded by a thin whitish halo [55,58,59,74,84,85,116] or structureless areas [42] - pinkish homogeneous structureless area or background, whitish scales or crusts, dotted and/or linear vessels [59,84,94] - polygonal structureless white and yellowish areas [84] - irregular linear parallel furrows “cracked riverbed-like” appearance [55]
Hailey–Hailey disease	- irregular whitish areas were separated by pink furrows (crumpled fabric or cloud pattern) [49,80], irregular combination of white and pink areas (cloud or iceberg pattern) [50,71] - polymorphous vessels predominantly in peripheral distribution, pink-whitish or pink-yellowish background, scales, erosions [83], red to brown linear ulcers with sharp angulated margins along with whitish macerated edges, pinkish-white background, peripheral arborizing telangiectasia [114] - diffuse white structureless areas and linear/linear-parallel erosions (tire-like appearance) [40]
Pseudoxanthoma elasticum	- multiple irregular yellowish areas alternating with prominent superficial linear vessels, yellowish areas may coalesce to form parallel strands [26,27] - distinct coalescing and reticulated yellow/white clods on a light purplish-red background [57,70,90] giving a cobblestone appearance [90] - yellow to ivory white non-follicular globules, the arrangement of dots, linear, broad, narrow mesh network, lines, and plaques on a pink or purplish-red background [46], and reticulated vessels [60] - yellowish-orange area with reddish and whitish areas [86] - yellowish-white structures coalescing into linear streaks, interspersed with erythema, exaggerated pigment network [51] - yellowish-brown structureless areas or background, semicircular, curved/serpiginous yellowish-brown lines, linear, dotted or hairpin vessels, keratotic plugs [67,101,102] - unspecific pattern of irregular pigmentation with a predominant yellowish-orange color alternating with reddish and whitish areas, microulcerations [81]
Fabry disease angiokeratoma	dark purple or red glomerular/lacunar/dotted/linear/irregular vascular structures with or without whitish veil [30]
Neurofibromatosis type 1 neurofibroma	pink/red homogeneous areas, peripheral pigment network, fissures, scar-like white areas in “star burst appearance” [40], peripheral pigmented network, fingerprint-like structures, peripheral halo of brown pigmentation, fissures, vessels [56]
café-au-lait macule	a homogenous brown pigmentation with perifollicular halo (face), reticular patterned brown pigmentation (neck) [77]
Tuberous sclerosis complex adenoma sebaceum (angiofibroma)	- multiple yellowish white globules or dots of varying length on brownish, reddish-brown, or pinkish-gray background [7,45,69] - dots of brown pigmentation [69] - bluish-white lacunae, red dots, and white globules [109]
ash leaf macule	white patch with irregular feathery border [7]
shagreen patch	yellowish globules, brownish background [7]
Basal cell nevus syndrome acral pits	- flesh-colored or pinkish irregular-shaped depressed lesions containing red globules in parallel lines [66,72,79,96,100] - blue structures and microarborizing vessels (more frequently seen in childhood) [66]
basal cell carcinoma	absence of pigment network, maple-leaf like structures, arborizing vessels, blue/grey ovoid nests, blue/grey globules and dots, concentric structures, spoke/wheel structures, and ulceration [61,66,72,73,79,96,100,105]
CYLD cutaneous syndrome trichoepithelioma	arborizing vessels, multiple milia-like cysts and rosettes, whitish background [41,82,92,103]
cylindroma and spiradenoma	- background pink coloration with ill-defined arborizing vessels and ill-defined blue structures [65,112] - white globules at the center [112] - absence of pigment network, white/ivory background, polymorphous vessels [99]
Noonan syndrome with multiple lentigines lentigines	pigment network, black dots or brown globules, branched streaks [44]
café noir spot (melanocytic nevi or lentigo simplex)	- pigment network, black dots, and dark globules [63] - branched streaks forming hyphae-like structures, light brown globules [44]

**Table 3 biomedicines-11-02717-t003:** Number of patients and number and localization of lesions analyzed according to different genodermatoses.

	Number of Patients	Number of Analyzed Areas or Lesions	Affected Areas
Dowling–Degos disease	1	3 areas	chest, back, axilla
Erythrokeratodermia variabilis et progressiva	2	6 areas	trunk, extremities
Monilethrix	2	15 trichoscopic fields of views	hair shaft
Noonan syndrome with multiple lentigines	3	154 lentigines 5 café noir spots	extremities, hands, trunk
CYLD cutaneous syndrome	3	12 trichoepitheliomas	scalp, face, shoulder
Fabry disease	3	37 angiokeratomas	neck, trunk, legs
Tuberous sclerosis complex	6	16 areas of adenoma sebaceum 4 ash leaf macules 2 shagreen patches	face, trunk, thighs
Pseudoxanthoma elasticum	7	14 areas	neck, axilla, cubital fossa
Darier disease	8	25 areas 7 nail findings	chest, back, neck, calf
Hailey–Hailey disease	14	38 areas 5 nail findings	axilla, sub-mammary, inguinae
Palmoplantar keratodermas	12	24 areas	palms, soles
Basal cell nevus syndrome	11	8 palmar pits 11 basal cell carcinomas	palms, soles, face, trunk
Neurofibromatosis type 1	20	45 neurofibromas 14 CALMS	trunk, extremities
Ichthyoses	27	59 areas	face, neck, trunk, extremities, palms

**Table 4 biomedicines-11-02717-t004:** Dermoscopic findings of genodermatoses following the methodology of Errichetti et al. [9].

Genodermatosis	Dermoscopic Findings				
	Vessels	Scales	Follicular Findings	Other Structures	Specific Clues
Ichthyosis vulgaris	-	fine white scales in criss-cross pattern (100%)	-	-	-
X-linked recessive ichthyosis	-	brown structures in rhomboid or mosaic with space in between (100%)	-	-	-
Autosomal recessive congenital ichthyoses (ARCI)					
Lamellar ichthyosis	dotted (50%)	quadrilateral brown structures with fine white scale around arranged in lamellar pattern (100%)	-	-	-
Congenital ichthyosiform erythroderma	dotted (100%)	diffuse white scales sometimes in rhomboid pattern (100%)	-	parallel white lines (100%)	erythema
Pleomorph ichthyosis	-	fine white scales in criss-cross pattern (100%)	-	-	-
Harlequin ichthyosis	dotted (100%)	yellow white scales in parallel pattern (100%)	-	-	excessive erythema
Dowling–Degos disease	dotted, linear curved (100%)	-	follicular plugs (100%)	yellow/ brown structureless areas (100%) white globules (100%)	-
Palmoplantar keratodermas Punctate	dotted (100%)	white (100%)	-	oval yellow areas, white lines (100%), brown dots (50%)	hyperkeratosis, fissures (100%)
Diffuse epidermolytic	erythematous edge: dotted (50%)	white (100%)	-	orange and yellow structureless areas, parallel or angulated white lines (100%), brown dots (12.5%)	hyperkeratosis, fissures, erythematous edge (100%)
Erythrokeratodermia variabilis et progressiva	dotted (100%)	fine white scales (100%) in rhomboid (25%) or criss-cross pattern (25%)	-	brown thick lines and structureless areas (100%) hyperkeratotic white globules (50%)	erythematous lines
Darier disease hyperkeratotic papules and plaques	dotted (48%), linear (48%)	yellowish scales/ crusts (72%)	-	parallel, perpendicular, and angulated lines (64%)	polygonal yellow/brown areas with whitish halo (100%) erosions (64%) erythema (100%)
pseudocomedones	-	-	follicular plugs (100%)	-	polygonal yellow/brown areas with whitish halo (100%)
Hailey–Hailey disease	dotted (68.42%) linear (52.63%)	white/yellow (50.00%)	-	white structureless areas (100%)	fissures, erosions (89.47%) livid parallel, perpendicular, or unspecifically arranged lines (89.47%)
Pseudoxanthoma elasticum	superficial linear (33.3%), reticulated (55.56%) or dotted (11.11%)	-	-	yellow/white globules (100%) that may coalesce into parallel (22.22%) or linear lines (22.22%), broad (11.11%) or narrow meshwork (22.22%) light purple (55.56%) or brown (44.44%) structureless areas	mild erythema (66.67%)
Tuberous sclerosis complex adenoma sebaceum (angiofibroma)	linear, linear curved (46.15%)	-	-	yellow/ white dots and globules, white structureless areas (100%), central brown dots surrounded by white circles (53.85%)	-
ash leaf macules	linear, linear curved (50%)	-	-	white structureless areas with feathery irregular border (50%), white globules coalescing into reticulated lines (50%)	-
shagreen patch	linear, linear curved, linear with branches (50%)	-	-	white/light yellow structureless areas (100%)	-

**Table 5 biomedicines-11-02717-t005:** Dermoscopic findings of genodermatoses following the standardized terminology of Kittler et al. [8].

Genodermatosis/Skin Manifestations	Dermoscopic Findings
Fabry disease	
angiokeratoma	- combination of reddish and purplish dots and globules and yellowish structureless areas covered by ***whitish veil***; globules are divided by yellow reticular lines (45.95%) - various sizes of dark blue and purplish dots and globules with ***whitish veil***, smaller dots and globules may be grouped (54.05%)
Neurofibromatosis type 1	
café-au-lait macules	structureless (homogenous) pigmentation with perifollicular hypopigmentation (73.33%) or reticular pattern of brownpigmentation (26.67%)
neurofibromas	pink/red structureless areas (100%), ***scar-like areas*** (97.8%), fissures (68.8%), ***fingerprint-like structures*** (80%), peripheral pigment network (37.8%), peripheral halo of brown pigmentation (57.8%)
Basal nevoid cell syndrome	
basal cell carcinoma	absence of pigment network (100%, ***maple-leaf like structures*** (63,64%), ***arborizing vessels*** (100%), ***blue/grey ovoid nests*** (81.82%), ***concentric structures*** (54.55%), ***spoke/wheel structures*** (45.45%), and ulceration (45.45%)
acral pits	flesh-colored (36.36%) or pinkish areas (63.64%) containing red dots in parallel lines (100%)
Noonan syndrome with multiple lentigines	
lentigines	- 1 to 3 mm in size, light brown to brown in color - homogenous light brown pigmentation - symmetric brown follicular pigmentation (***pseudonetwork***) (100%)
café noir spots	- symmetric, in certain areas irregular brown follicular pigmentation (***pseudonetwork***) (100%) - brown pigmentation in a ***cobblestone-like pattern*** (brownish polygonal large clods) (20%)
CYLD cutaneous syndrome	
trichoepithelioma	***milia-like cysts***, pinkish/whitish background, ***arborizing vessels*** (100%)

Standardized metaphoric terms are in bold and italics.

**Table 6 biomedicines-11-02717-t006:** Trichoscopic and onychoscopic findings of genodermatoses.

Genodermatosis	Trichoscopic or Onychoscopic Findings	Our Findings
Monilethrix	regular constrictions of the shaft with elliptical nodes separated by internodes [64,76,87,107], regularly bent ribbon sign [7,43,88,89,93] or beaded appearance [48,106] rosary beads with nodes and constrictions [54] irregular atypical beads [108]	100% (2 patients)
Darier disease	reddish/white longitudinal nail bands with a V-shaped nick at the free margin [55]	87.5% (7 patients)
Hailey-Hailey disease	longitudinal white bands [23,24,49]	35.71% (5 patients)
Tuberous sclerosis complex		
subungual red comets	tortuous or corkscrew-like	0%
	vessels with a narrow proximal tail and a dilated distal head, surrounded by a whitish halo, parallel binary tortuous capillaries [68,91]	(0 patients)

## Data Availability

The data that support the findings of this study are available upon reasonable request from the corresponding author.

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
