# Peer review of "Dermoscopic Patterns of Genodermatoses: A Comprehensive Analysis"

_biomedicines, 2023, doi:10.3390/biomedicines11102717_

Round 1
Reviewer 1 Report
The abstract presents the article appropriately, giving a clear idea of what to expect in the following sections.
The introduction is not very fluent in the initial part, it can be summarized.
On line 61 after the word “xerosis” there is a typo (a period that should not be there).
The “materials and methods” section is complete and detailed.
“Figure 1” should be enlarged because it is difficult to read.
The dermoscopic images are well described in the captions.
Grammar and vocabulary are correct, the language is adequate; however in some parts the concepts could be expressed in a simpler way
Author Response
We would like to thank you for your precise review and suggestions, we listed our responses to your comments below.
The introduction is not very fluent in the initial part, it can be summarized.
Grammar and vocabulary are correct, the language is adequate; however in some parts the concepts could be expressed in a simpler way.
We thank the reviewer for this suggestion, we summarized and simplified some sentences in the initial part of the introduction. „Genodermatoses are a clinically and genetically heterogenous group of inherited skin disorders. These are chronic conditions that present with variable severity of dermatological symptoms and may be associated with extracutaneous manifestations that can have a severe impact on the overall health and quality of life of patients. Diagnosing inherited skin diseases is difficult because these conditions are both rare and diverse. The multistep diagnostic algorithm for inherited skin diseases suggests considering phenotypic features and clinical data, mode of inheritance, target proteins, and genetic variants in the diagnosis of genodermatoses”.
On line 61 after the word “xerosis” there is a typo (a period that should not be there).
We thank for this careful observation, the typo from line 61 was removed.
“Figure 1” should be enlarged because it is difficult to read.
We enlarged Figure 1. Unfortunately this process was limited in order to keep the margins, but we hope you fill find the new figure adequate.
Reviewer 2 Report
I enjoyed reviewing the article "Dermoscopic Patterns of genodermatoses: a comprehensive analysis" by Plázár D. et al.
The manuscript is globally well-written. The introduction section is exhaustive, listing the main known genodermatoses. Materials and methods are correctly described. However, it is still being determined if patients included in the study were already diagnosed with genodermatoses or if the diagnosis has been made after the survey. In the results section, the multiple figures panels well represent the diseases. However, the authors' data need to be more underlined, especially regarding the value of dermoscopy in the follow-up of genodermatoses.
Author Response
We would like to thank you for your valuable insight, and we hope that you find the modified version of the manuscript is worthy of consideration as well.
However, it is still being determined if patients included in the study were already diagnosed with genodermatoses or if the diagnosis has been made after the survey.
We clarified that patients included the study were previously diagnosed by histological/ molecular genetical verification. We corrected the sentence in lines 218-219: ’A total of 119 patients with 14 different inherited disorders were evaluated. Patients with the previously established diagnosis of genodermatosis were included.’
In the results section, the multiple figures panels well represent the diseases. However, the authors' data need to be more underlined, especially regarding the value of dermoscopy in the follow-up of genodermatoses.
We also underlined our datas regarding the value of dermoscopy in the follow-up in lines 394-403: ’Dermoscopy may also enhance monitoring of disease activity and accurate follow-up of treatment response. Errichetti et al. successfully used dermoscopy in psoriasis. According to their results, it was useful for following therapy response, detecting steroid-induced skin atrophy by visualizing characteristic linear vessels and disease recurrence. In our cases, steroid-induced skin atrophy could be seen in patients with HHD and DD. In addition, with the use of dermoscopy, we monitor the efficiency of topical therapy for adenoma sebaceum (angiofibroma) in TSC. In our clinical practice, we use dermoscopy for the follow-up of patients with BCNS or NSML to detect potential skin tumors.’
Reviewer 3 Report
Dear Author,
I have read your work with interest. I find your revision to be very captivating and well-written. The images are exceptional.
Sincerely,
Dear editor,
I believe this article is very intriguing, interesting, and well-presented.
Sincerely,
Author Response
We would like to thank you for the positive feedback, we sincerely appreciate the review. We are glad that you find our work worthy of consideration.